# Risk of Developing Epilepsy after Autoimmune Encephalitis

**DOI:** 10.3390/brainsci11091182

**Published:** 2021-09-08

**Authors:** Ariadna Gifreu, Mercè Falip, Jacint Sala-Padró, Neus Mongay, Francisco Morandeira, Ángels Camins, Pablo Naval-Baudin, Misericordia Veciana, Montserrat Fernández, Jordi Pedro, Belia Garcia, Pablo Arroyo, Marta Simó

**Affiliations:** 1Epilepsy Unit, Neurology Service, Hospital Universitari de Bellvitge, Hospitalet de Llobregat, 08907 Barcelona, Spain; agifreu@bellvitgehospital.cat (A.G.); jsalap@bellvitgehospital.cat (J.S.-P.); 2Neurology Ward Unit, Neurology Service, Hospital Universitari de Bellvitge, Hospitalet de Llobregat, 08907 Barcelona, Spain; smongay@bellvitgehospital.cat (N.M.); pablo.arroyo@bellvitgehospital.cat (P.A.); 3Laboratory Service, Immunology Department, Hospital Universitari de Bellvitge, Hospitalet de Llobregat, 08907 Barcelona, Spain; fmrego@bellvitgehospital.cat; 4MRI Unit, IDI (Institute of Image Diagnosis), Hospital Universitari de Bellvitge-IDIBELL, Hospitalet de Llobregat, 08907 Barcelona, Spain; acamins@bellvitgehospital.cat (Á.C.); pnaval@bellvitgehospital.cat (P.N.-B.); montse.fernandez@idi.gencat.cat (M.F.); 5Neurophysiology Department, Neurology Service, Hospital Universitari de Bellvitge, Hospitalet de Llobregat, 08907 Barcelona, Spain; mveciana@bellvitgehospital.cat (M.V.); jpedro@bellvitgehospital.cat (J.P.); belia.garcia@bellvitgehospital.cat (B.G.); 6Neuro-Oncology Unit, Hospital Universitari de Bellvitge-ICO L’Hospitalet (IDIBELL), Hospitalet de Llobregat, 08907 Barcelona, Spain; msimo@bellvitgehospital.cat

**Keywords:** autoimmune encephalitis, autoimmune-related epilepsy, acute symptomatic seizures, acute symptomatic seizures related to autoimmune encephalitis, hippocampal atrophy, immunotherapy, interictal epileptiform discharges

## Abstract

Background: Acute symptomatic seizures (ASS) are a common manifestation of autoimmune encephalitis (AE), but the risk of developing epilepsy as a sequela of AE remains unknown, and factors predisposing the development of epilepsy have not been fully identified. Objective: To assess the risk of developing epilepsy in AE and study related risk factors. Materials and methods: This was a retrospective single centre study including patients diagnosed with AE according to criteria described by Graus et al., with a minimum follow-up of 12 months after AE resolution. The sample was divided according to whether patients developed epilepsy or not. Results: A total of 19 patients were included; 3 (15.8%) had AE with intracellular antibodies, 9 (47.4%) with extracellular antibodies, and 7 (36.8%) were seronegative. During follow-up, 3 patients (15.8%) died, 4 (21.1%) presented relapses of AE, and 11 (57.89%) developed epilepsy. There was a significant association between the development of epilepsy and the presence of hippocampal atrophy in control brain magnetic resonance imaging (MRI) (*p* = 0.037), interictal epileptiform discharges (IED) on control electroencephalogram (EEG) (*p* = 0.045), and immunotherapy delay (*p* = 0.016). Conclusions: Hippocampal atrophy in neuroimaging, IED on EEG during follow-up, and immunotherapy delay could be predictors of the development of epilepsy in patients with AE.

## 1. Introduction

Autoimmune encephalitis (AE) is an inflammatory autoimmune disorder of the brain, which produces a rapidly progressive encephalopathy, usually evolving over weeks to months. The clinical features consist of subacute onset of neuropsychiatric symptoms such as behavioural disturbance, psychosis, focal seizures, cognitive and memory dysfunction, abnormal movements, dysautonomia, and decreased level of consciousness [1].

Acute symptomatic seizures (ASS) are a common, well-recognised, and often prominent manifestation of AE [2,3,4], occurring during the acute or early stages of the disease in the majority of patients [3]. Recently, the definition of acute symptomatic seizures related to AE (ASAE) was proposed to differentiate the condition from a chronic condition called autoimmune-related epilepsy (ARE). ASAE are seizures occurring in the setting of the active phase of immune-mediated encephalitis, initially or in relapse, which sometimes take weeks or even months to resolve [4]. The determination of active disease in AE generally is achieved by a combination of laboratory findings (elevated titres of antibodies to surface antigens, cerebrospinal fluid [CSF], and imaging markers of inflammation) and clinical evidence of active encephalitis [4]. There is no strict operational time definition for these disorders given the wide spectrum of clinical presentation, which can vary according to the particular associated antibody (Ab) and the timing of immune-targeted therapy [4].

By contrast, ARE refers to a chronic situation determined to be secondary to autoimmune brain diseases, with an enduring predisposition to unprovoked seizures [4]. The development of epilepsy in such cases may be the consequence of two processes: It may result from an ongoing active immune-mediated inflammatory process or autoimmune brain disease that persists after the acute phase of AE, or from associated irreversible immune-mediated structural brain injury, such as hippocampal atrophy or multifocal cortical cell loss (non-inflammation dependent), or a combination of the two processes, which alters neuronal networks and persists after the inflammatory process resolves [2,3,4]. Therefore, although neuroinflammation may trigger ASS, this does not necessarily imply that the subsequent development of chronic seizures is inflammatory-dependent [3].

The risk of developing epilepsy as a sequela of AE remains unknown [3]. Overall, AE is often immunotherapy-responsive and appears to have a low risk for epilepsy [3]. ASS usually shows good response to immunotherapy [2], and, in more than 70% of patients with AE, the associated seizures are successfully treated with immunotherapy and antiseizure medication (ASM), with most not requiring chronic treatment [3]. Nevertheless, in a proportion of cases, seizures can persist over time despite adequate immunotherapy regimens and in the absence of clear evidence of active inflammation, following resolution of the acute phase of AE [2,4], and can finally evolve to a chronic situation defined as epilepsy.

Although seizures are frequent in all types of AE, the risk for epilepsy varies according to the target autoantigen, being lower in AE with antibodies (Ab) against structures of the neuronal cell-surface antigens (NSA) than in AE with Ab against intracellular antigens [3]. In the first case, after the encephalitis is successfully treated, the risk of developing epilepsy is low (<15%), and the response to immunotherapy is substantially better than in those with central nervous system (CNS) disorders with Ab against intracellular proteins, which appear to be related to T cell-mediated mechanisms, such as glutamic acid decarboxylase (GAD) antibody-associated epilepsy, which are much less responsive to immunotherapy [3,4].

Given the complexity of the pathogenesis of AE and its phenotypical spectrum, factors predisposing the development of epilepsy in AE have not yet been fully identified.

## 2. Objective

To assess the risk of developing epilepsy in AE and study the related risk factors.

## 3. Materials and Methods

### 3.1. Study Design, Patients, and Inclusion/Exclusion Criteria

This was a retrospective study including a cohort of patients diagnosed with AE according to criteria described by Graus [1] and referred to the Epilepsy Unit of the Hospital Universitari de Bellvitge (HUB; Barcelona, Spain) from 2000–2020 with a minimum follow-up of 12 months in our outpatient clinic after the resolution of AE. The patients included fulfilled the criteria of definitive or possible AE. All patients had undergone an electroencephalogram (EEG) and brain magnetic resonance imaging (MRI) in the acute phase and most underwent control EEG and brain MRI during follow-up.

The sample was divided according to whether the patients developed epilepsy or not, defined as >1 unprovoked seizure after resolution of the acute phase. The acute phase was defined both clinically by the presence of symptoms suggestive of encephalopathy such as altered consciousness, altered attention, presence of hallucinations or delusions, altered circadian rhythm, among others and by paraclinical tests, considering the period when brain inflammation could be observed (determined by EEG, CSF study or neuroimaging techniques: MRI or Fluorine-18-fluorodeoxyglucose positron emission tomography/computed tomography—^18^F-FDG PET/CT). When two paraclinical tests were carried out separated by months and the first was pathologic (^18^F-FDG PET/CT showing hypermetabolism, CSF with inflammatory findings, and EEG without normal background) and the second was normal or showed important improvement, clinical changes between the two results marked the end of the acute phase. In the case of presenting remote seizures after resolution of the acute phase of AE, complementary examinations should be carried out to differentiate ARE from ASS in the context of a recurrence of AE. These complementary studies should include a CSF study, a brain MRI, and even a brain ^18^F-FDG PET/CT if available, in order to determine the presence of inflammatory activity at that time, which could correspond to a relapse of AE. EEG monitoring would not be strictly necessary to differentiate a relapse of AE or an ARE, but rather would help in decision making regarding the adjustment of treatment with ASM.

Risk factors associated with the development of epilepsy were analysed.

The aetiologies that must be excluded when diagnosing probable ASAE or ARE are described in Graus et al. [1] and Falip et al. [5]. Seizure classification was made according to the International League Against Epilepsy 2017 operational classification of seizure types [6].

The study was approved by the Ethical Committee of the HUB (PI10/00738). Written informed consent was obtained from all patients.

### 3.2. Variables

The demographic characteristics of the patients were collected, including the mean age at AE onset and gender; clinical manifestations such as seizure semiology, cognitive impairment, onset as status epilepticus; AE relapses, deaths, serum, and CSF results (whenever possible) detecting Ab against intracellular antigens or NSA; brain MRI features at baseline and in follow-up with special attention to hippocampal atrophy; EEG pattern and detection of interictal epileptiform discharges (IED) or ictal epileptiform activity in the initial and the control EEG; immunotherapy received (classified as first- or second line-treatment), and the delay until treatment was administered.

Both control MRIs and EEGs were carefully examined by two independent epileptologists, evaluating shrinkage of the volume in both hippocampi and IED.

### 3.3. Organ-Specific Immunological Analysis

All Ab were measured in serum and CSF (when available). This analysis included onconeural Ab, Ab against NSA, and GAD Ab. Onconeural Ab (Hu, Yo, Ma, Tr, amphyphysin, CV2/CRMP5, SOX-1, Zic4) testing was conducted in the HUB using immunoblotting. NSA Ab included LGI1/contactin-2-associated protein (CASPR2), N-methyl-D-aspartate receptor (NMDAR), α-amino-3-hydroxy-5-methyl-4-isoxazolepropionic acid receptor (AMPAR), γ-aminobutyric acid receptor (GABABR), and dipeptidyl-peptidase-like protein-6 (DPPX). These specific immunological profiles were measured at the Neuroimmunology Unit of the Hospital Clinic from 2010 to 2018. In 2018, NSA Ab were analysed with indirect immunofluorescence in cells transfected with the antigen using the commercial kit IIFT Autoimmune Encephalitis Mosaic 6, Euroimmun in HUB. All positive neuronal surface antibodies were sent to the Hospital Clinic and were identified by immunocytochemistry of rat hippocampal neuronal cultures and confirmed in a cell-base assay, as described elsewhere [7]. GAD65 Ab were analysed as described elsewhere [7], while GAD positivity was considered for serum titres above 1000 IU/mL.

The rest of the methodology employed in the study as well as the neuroimaging and EEG studies have been described previously [7].

### 3.4. Statistical Analysis

Statistical significance for intergroup differences was assessed with the Pearson’s chi-square or Fisher’s exact text for categorical variables, and the Student’s t or Mann–Whitney U test for quantitative variables. A survival analysis (Cox proportional-hazards model) considering the development of epilepsy over time and its related factors was also performed. A *p* value < 0.05 was considered statistically significant.

## 4. Results

### 4.1. Clinical and Demographic Characteristics

A total of 23 patients were initially recruited. One patient was excluded because the diagnosis of AE was doubtful: while presenting with ASS and the presence of a paraneoplastic Ab (anti-Zic4) was detected, the patient did not meet the diagnostic criteria for AE because of a lack of clinical features or brain MRI changes suggestive of AE. Thus, the final number of patients recruited was 22. It was not possible to determine whether or not 3 patients would have developed epilepsy as they died in the acute phase before 12 months of follow-up. Therefore, according to our inclusion criteria and because these patients could not be classified according to the outcome of epilepsy, they were excluded from the statistical analysis, and finally 19 patients were included.

The clinical and demographic characteristics of the study cohort are presented in Table 1. MRI findings at onset more frequently showed unilateral limbic system oedema or hyperintensity (47.37%). Cognitive impairment was detected in 63.16% of patients during follow-up. Nonetheless, it should be mentioned that standardised collection of the outcome of cognitive impairment was not performed. The main limitation was that not all patients underwent a neuropsychological examination after AE, and some were classified based on the clinical information collected in the medical history under the subjective opinion of the treating neurologist. Besides, the modified Rankin Scale score at both onset and during follow-up might be more informative about disease severity and evolution, but the lack of standardised data collection makes it impossible to describe this characteristic of our cohort. Thirteen patients continued with treatment with ASM during follow-up. Half of the patients (50%) who did not develop epilepsy continued to receive treatment with one ASM, while all the survivors who developed epilepsy remained under ASM treatment.

Over the course of the follow-up, 3 patients (15.79%) died and 4 presented relapses of AE (21.05%). Special attention should be paid to causes of death, since this is a relatively uncommon outcome of AE. Among the 3 excluded patients, in two cases, the causes of death in the acute phase of AE were non-convulsive status epilepticus (in one of the patients in whom no Ab was detected, death occurred during a relapse, while in the other case in which an anti-GAD Ab was detected, death occurred in the debut). In the third case (in which an anti-NMDA Ab was detected), the cause of death was a spontaneous intraparenchymal haematoma with intracranial hypertension in the context of an active malignant neoplastic process (small cell pulmonary carcinoma).

Along the follow-up, 11 out of 19 patients (57.89%) developed epilepsy after the acute phase, while the remaining 8 did not (42.11%). Among these 11 patients, 7 (63.64%) continued having seizures after the debut of AE without a free interval period; 6 continue having seizures to date; and 1 had seizures during 13 years after the debut of AE and stopped having seizures 1 year after initiation of immunomodulatory treatment. The presence of Ab in these cases was variable: 3 anti-GAD, 1 anti-VGKC, and 3 seronegative. On the other hand, among the 11 patients who developed epilepsy, 4 (36.36%) did so after a variable free interval period after the debut of AE (Figure 1). In these patients, the Ab detected were 2 anti-NMDA and 2 anti-LGI1.

The average time from resolution of AE to the first seizure was 13.09 months, with a very wide range (from 0 up to 79 months). Among patients who developed epilepsy, 8 (72.73%) did so during the first 12 months of follow-up, and 9 (81.82%) during the first 24 months. The risk of developing epilepsy was 42.11% at 12 months of follow-up, 47.37% at 24 months, and 57.89% at the last follow-up.

### 4.2. Risk Factors for Developing Epilepsy

The clinical and paraclinical characteristics of patients who developed epilepsy during the follow-up were compared with those who did not (Table 2).

In the univariate analysis, there was a statistically significant association between developing epilepsy and the presence of hippocampal atrophy in successive brain MRI scans (*p* = 0.037), and the presence of IED in control EEG (*p* = 0.045) (Table 2, Figure 2).

Moreover, an association between immunotherapy delay and development of epilepsy was found, demonstrating an increased risk of developing epilepsy in patients with higher latency until the initiation of immunotherapy (*p* = 0.016) (Table 2). The remaining variables analysed did not show statistically significant associations with epilepsy.

The presence of anti-GAD Ab was statistically significantly related to the absence of a seizure-free period in patients developing epilepsy after AE resolution (42.86% vs. 0%, *p* = 0.040). No relationship was found with the other Ab.

In the survival analysis, no factor was found to be related with developing epilepsy over time, although the presence of IED in the control EEG showed a statistical trend to significance (odds ratio 4.955–confidence interval 0.988–24.848; *p* = 0.052).

## 5. Discussion

In our series, the risk of developing epilepsy after an AE was 42.11% at 12 months, similar to other recent studies [2,8] and higher than others [3,9]. Among patients who developed epilepsy, most (72.73%) did so in the first 12 months of follow-up, although, in one patient, the epilepsy debut occurred after 7 years.

The predisposing factors associated with the risk of developing epilepsy included hippocampal atrophy at follow-up, the presence of IED in the control EEG, and a delay in initiating immunotherapy. Another study also observed that IED in control EEG increases the risk of developing epilepsy [9]. On the other hand, one study [10] observed progressive hippocampal atrophy in NMDAR and LGI1 encephalitis, which was related to a decline in memory but not to the risk of developing epilepsy; however, this was not the aim of that study. Many other authors have observed a relationship between the risk of developing epilepsy and a delay in diagnosis or the initiation of immunotherapy, sometimes because of atypical and insidious symptoms [2,9,11,12]. In addition, an association between the presence of anti-GAD Ab and the absence of a seizure-free period in patients developing epilepsy was found.

According to the literature available [3], other risk factors associated with the development of epilepsy are neuronal Ab subtype and debut as status epilepticus. In our opinion, both may be reasonable risk factors, but could not be demonstrated in our study because of our sample characteristics; for instance, in our series, 2 patients who presented with status epilepticus died in the acute phase, and thus, had to be excluded.

The risk of developing epilepsy after presenting ASS also depends on the aetiology. When comparing its risk with other aetiologies, AE carries a greater risk of developing epilepsy during long-term follow-up. In the case of infections of the CNS, epilepsy occurs in 6.8 to 8.3% of cases, and risk factors associated with the development of epilepsy are having ASS and certain infectious agents [13]. Stroke is the most common cause of seizures and epilepsy in the elderly population [14,15,16], and the incidence of post-stroke epilepsy varies widely from 2 to 20% [16,17,18,19,20,21,22,23,24,25]. Major risk factors are haemorrhagic strokes (10.6–15.4%) [26,27,28,29,30,31,32,33] and the time interval of seizure onset following a stroke [16,34,35]. In addition, epilepsy is common in traumatic brain injury, which may arise after many years [36]. Its incidence ranges from 1.9% to over 30%, depending on the severity of trauma and the duration of follow-up [36,37,38,39]. Intracerebral haemorrhage (mainly subarachnoid and subdural) carries an increased risk [36]. In summary, the risk of developing epilepsy after an AE seems the highest among aetiologies that produce brain injury.

Finally, it should be noted that the acute phase of AE is difficult to define due to the scarce evidence available in this regard as well as the limited use of complementary tests, which are often not sufficiently sensitive or specific or cannot be performed frequently due to limitations in cost, organisational capacity, or tolerance. In these cases, definition of the acute phase mainly relies on comprehensive evaluation of the clinical evolution of the patient.

## 6. Limitations

The present study has some methodological limitations such as the small sample size, its retrospective nature (which may have missed some important information about the cognitive and functional evaluation of these patients because data were not collected in a standardised way and some data were missing), the limited number of EEG exams available during follow-up, the fact that some patients who did not develop epilepsy continued to be treated with ASM due to patient fear of suffering new seizures or new status epilepticus, and a relatively short follow-up period (which may have excluded patients who have not yet developed epilepsy). A possible sampling bias should also be considered, as patients who developed epilepsy were more likely to be followed at our outpatient clinic. Considering these limitations, further prospective multicentre studies with larger sample sizes should be carried out to confirm our findings and draw definitive conclusions.

## 7. Conclusions

Epilepsy after AE is common. Hippocampal atrophy in neuroimaging, IED on EEG during follow-up, and a delay in initiating immunotherapy could be predictors of the development of epilepsy in patients with AE, suggesting that it is important to perform sequential neuroimaging studies and EEG during the follow-up of patients with AE and to start treatment early to improve the prognosis of these patients.

## Figures and Tables

**Figure 1 brainsci-11-01182-f001:**
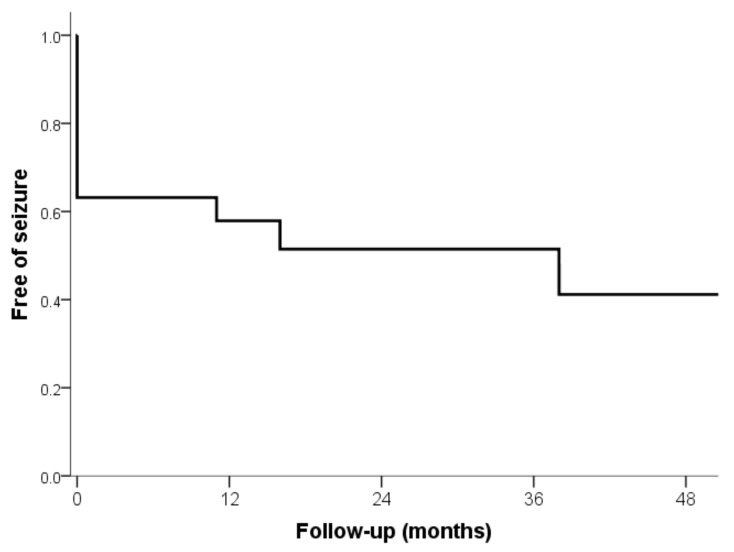
Survival analysis (Kaplan–Meier): time of onset of the first seizure after the resolution of the AE during follow-up.

**Figure 2 brainsci-11-01182-f002:**
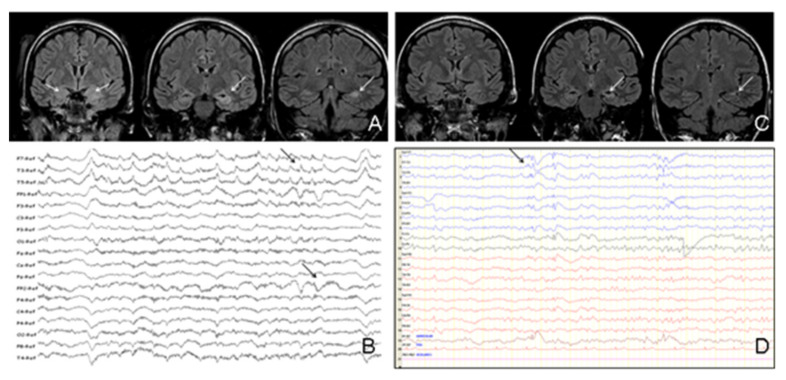
(**A**) Initial brain MRI of one patient with AE due to NMDAR Ab (acute phase). Coronal fluid-attenuated inversion recovery (FLAIR) sequences showing bilateral amygdalar and left hippocampus hyperintensities and increased volume of both structures (white arrows). (**B**) Initial EEG of the same patient (acute phase). Reference montage showing IED over the left frontotemporal region and generalised beta activity (black arrows). (**C**) Control brain MRI of the same patient after resolution of AE. Coronal FLAIR sequences showing persistence of hyperintensity in left hippocampus and development of hippocampal atrophy (mainly in the tail) (white arrows). (**D**) Control EEG of the same patient after resolution of AE. Bipolar montage showing IED over the left frontotemporal region (black arrows).

**Table 1 brainsci-11-01182-t001:** Clinical and demographic characteristics of the study cohort.

Variables	Number (%)	Mean (SD)	Range
Patients	19		
Gender	M 10 (52.63)		
	F 9 (47.37)		
Age (years)		52.79 (20.29)	20–85
Average age at onset (years old)		44.21 (18.03)	18–69
Follow-up (years)		7.16 (6.15)	1–18
Deaths	3 (15.79)		
Debut as status epilepticus	8 (42.11)		
Main initial seizure type (each patient can present different types):			
BTCS	4 (21.05)		
FAS	3 (15.79)		
FIAS	12 (63.16)		
FBDS	2 (10.53)		
Aphasic seizure	1 (5.26)		
Main seizure type during follow-up:			
BTCS	6 (31.58)		
FAS	3 (15.79)		
FIAS	6 (31.58)		
FBDS	1 (5.26)		
Aphasic seizure	0 (0)		
Cognitive impairment	12 (63.16)		
AE relapse	4 (21.05)		
Presence of Ab (serum or CSF)	12 (63.16)		
Intracellular Ab	3 (15.79)		
Extracellular Ab	9 (47.37)		
Type of Ab:			
Anti-GAD	3 (15.79)		
Anti-NMDAR	5 (26.32)		
Anti-LGI1 or VGKC	3 (15.79)		
Paraneoplastic (anti-Yo)	1 (5.26)		
Pathological initial brain MRI (the third one can overlap):	16 (84.21)		
Unilateral limbic system oedema/hyperintensity	9 (47.37)		
Bilateral limbic system oedema/hyperintensity	6 (31.58)		
Extra-limbic system oedema/hyperintensity	3 (15.79)		
Control brain MRI with hippocampal atrophy	8 (42.11)		
Epileptiform initial EEG or ictal EEG findings	13/18 (72.22)		
Control EEG with IED	7/12 (58.33)		
Pathological initial ^18^F-FDG PET/CT	10/14 (71.43)		
Control ^18^F-FDG PET/CT with temporal hypometabolism	8/14 (57.14)		
First-line immunotherapy	16 (84.21)		
Second-line immunotherapy	8 (42.11)		
Time until the start of immunotherapy (months)		1.5 (0.75–28.5) *	0.10–144
Treatment with ASM during follow-up	13 (68.42)		

BTCS: bilateral tonic–clonic seizures; FAS: focal aware seizure; FIAS: focal impaired awareness seizure; FBDS: facio-brachio-dystonic seizures; AE: autoimmune encephalitis; Ab: antibodies; CSF: cerebrospinal fluid; GAD: glutamic acid decarboxylase; NMDAR: N-methyl-D-aspartate receptor; LGI1: leucine-rich glioma inactivated 1; VGKC: voltage-gated potassium channel; MRI: magnetic resonance imaging; EEG: electroencephalogram; IED: interictal epileptiform discharges; ^18^F-FDG PET/CT: Fluorine-18-fluorodeoxyglucose positron emission tomography/computed tomography. * median (IQR).

**Table 2 brainsci-11-01182-t002:** Comparison of clinical and paraclinical variables of patients who developed epilepsy and those who did not.

Variables	Epilepsy (11/19) (57.89%)	No Epilepsy (8/19) (42.11%)	*p* Value
Gender	M 6/11, F 5/11	M 4/8, F 4/8	NS
Average age at onset	48.64 years old	38.13 years old	NS
Debut as status epilepticus	5/11	3/8	NS
AE relapse	2/11	2/8	NS
Presence of Ab (serum or CSF)	8/11	4/8	NS
Intracellular Ab	3/11	0/8	NS
Extracellular Ab	5/11	4/8	NS
Pathological initial brain MRI	10/11	6/8	NS
Control brain MRI with hippocampal atrophy	7/11	1/8	0.037
Epileptiform initial EEG or ictal EEG findings	8/11	5/8	NS
Control EEG with IED	6/7	1/5	0.045
Pathological initial 18F-FDG PET/CT	4/6	6/8	NS
Control 18F-FDG PET/CT with temporal hypometabolism	5/8	3/6	NS
First-line immunotherapy	9/11	7/8	NS
Second-line immunotherapy	4/11	4/8	NS
Time until the start of immunotherapy	6 (2.5–37) * months	1 (0.66–1) * months	0.016
Treatment with ASM during follow-up	9/11	4/8	NS

AE: autoimmune encephalitis; Ab: antibodies; CSF: cerebrospinal fluid; MRI: magnetic resonance imaging; EEG: electroencephalogram; IED: interictal epileptiform discharges; 18F-FDG PET/CT: Fluorine-18-fluorodeoxyglucose positron emission tomography/computed tomography; ASM: antiseizure medication. NS: non-significant. * median (IQR).

## Data Availability

Data that support the findings of this study are available from the corresponding author upon reasonable request.

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
