# Peer review of "Risk of Developing Epilepsy after Autoimmune Encephalitis"

_brainsci, 2021, doi:10.3390/brainsci11091182_

Round 1

Reviewer 1 Report

In this paper the Authors retrospectively analyzed a cohort of patients diagnosed with autoimmune encephalitis (AE) focusing on the subgroup later developing autoimmune-related epilepsy (ARE): they found MRI-detected hippocampal atrophy and interictal epileptiform discharges (IEDs) in follow-up EEG exams to be the only findings associated with ARE, whereas the diagnostic /therapeutic delay did not appear to significantly affect the outcome, contrary to some previous studies.

The topic of autoimmune-associated epilepsy is crucial in the context of AE long-term management and prognosis, and the Authors’ attempt to cast light on its risk factors is noteworthy, despite the small sample size and the heterogeneity of the study population. The introduction is well written and exhaustively illustrates the differences – in terms of significance and possible underlying pathophysiological mechanisms – between seizures arising during AE acute stage and follow-up, respectively. Nevertheless, the rest of the manuscript is quite succinct and poor in electro-clinical details, which likely reflects an authorial choice, but deserves some rethinking. In particular, I have some concerns and observations which I would like the Authors to comment on:

  1. In the Methods section, the Authors state that only patients with a minimum follow-up of 12 months after AE resolution were included. However, 3 of the 23 enrolled subjects died in the acute phase, before 12 months of follow-up, which contradicts the inclusion criteria, not to mention the objective of the study itself. I think these subjects should be excluded from the initial cohort. Besides, in the results the Authors report that 11 out of 23 participants developed epilepsy, 8 did not and 3 died before the resolution of the acute phase: if I am not mistaken, the sum of these subgroups account to 22 instead of 23 (as also shown in Table 2). These data should be revised, or else clarified.
  2. “Death” after the acute stage (important point: was it related to an underlying malignancy?) was selected as one of the variables considered for statistical analysis. However, the number of deaths is fortunately limited (3), and it might be argued –as a general principle- that death during follow-up could “prevent” the development of epilepsy, considering the possible long duration of the latency period. I suggest the Authors analyze the correlation between ARE and the modified Rankin Scale (mRS) score at both AE onset and during follow-up (instead of death), which might be more informative about disease severity and evolution.
  3. As illustrated in the Methods, the Authors considered the acute stage of AE as the period characterized by evidence of brain inflammation based on CSF analysis and neuroimaging findings. However, CSF analysis and MRI scans are unremarkable in up to 50% of AE cases, while in other patients typical MRI-detected alterations (e.g. increased T2/FLAIR signal of the mesial temporal structures) can persist in time. Besides, in daily practice these laboratory/neuroimaging exams (including PET studies) cannot be performed at short intervals due to costs/organizational difficulties and/or tolerability (as in the case of lumbar puncture). Therefore , the definition of the acute phase can be very challenging at times, as correctly stated in the introduction, and it mainly relies on the comprehensive evaluation of the patient’s clinical evolution. In light of this, the Authors should consider slightly revising the definition of the acute stage of AE provided in the Methods, and mentioning this issue in the discussion.
  4. For completeness, some details about the patients’ characteristics should be added in the text –or at least in the table – although they were not considered for analysis: e.g. seizure type in the early stage and during follow-up (indeed, in a previous study by Steriade et al, GTCS at AE onset were associated with ARE), mRS score, MRI findings at onset, cognitive impairment. In particular, the treatment with ASMs in the study cohort (indicated in Table 2) should be further clarified in the Results section, and discussed as a possible confounding/interfering factor.
  5. In this work, the therapeutic delay did not appear significantly correlated with ARE, although a trend was observed. I find absolutely reasonable (even likely) that the therapeutic delay affects the epilepsy outcome in these patients – as the Authors suggested; however, the “large difference in the absolute numbers” between epileptic and non epileptic patients should be less emphasized, or interpreted with caution, considering that the time to IT start ranged from 0.1 to 144 months: indeed, this means that one participant was treated after 12 years from AE onset, which might determine a bias.
  6. As previously stated, the discussion appears a little “off-balance”, given the long dissertation about the risk of developing epilepsy after various brain injuries. The Authors might consider revising it, taking into account the suggestions made at 3) and 4).
  7. The limited number of EEG exams available during follow-up should be mentioned among the study limitations.
  8. The tables are not properly formatted (in Table 2, the top 9 lines are also repeated at the bottom the table). Please, revise.
  9. The term “critical initial EEG” (if referring to the detection of seizures during the EEG exam) should be changed with “ictal EEG findings”

Reviewer 2 Report

The Authors report the interesting subject of the persistence or recrudescence of epileptic seizures following acute autoimmune encephalitis. The topic is interesting but needs clarification on some points.

The Authors rightly make a distinction between acute symptomatic seizures related to autoimmune encephalitis (ASAE), and epilepsy that occurs at a distance from the acute phase, called autoimmune-related epilepsy. What kind of diagnostic work-up do the Authors suggest in case of remote seizures from the end of an autoimmune encephalitis? Tests such as cerebrospinal fluid, Magnetic Resonance, EEG monitoring should be performed to exclude e relapsing of autoimmune encephalitis? Can Authors better specify which diagnostic work-up is needed for differential diagnosis between acute and chronic forms?

It is not entirely clear how the authors consider the “resolution of the acute phase”.  This is particularly true for that autoimmune encephalitis that have a strong psychiatric expression, which often lasts well beyond the acute stage of the disease.  Should a time criterion be included in the definition of acute phase resolution? And if yes, how many months? Can the Authors specify on which clinical elements, the acute phase resolution diagnosis, is based?

Regarding the time between the resolution of the AE and the first seizure, it would be interesting to have a graph with the different times of onset of the first seizure after the acute phase. For example, it would be interesting to know how many patients without free interval from the acute phase of the disease (time 0 days) were. Moreover, could be very interesting to know if the patients without free interval were correlated to a specific autoantibody.

Unless we can draw a very clear line between the acute phase and the outcomes phase, it would be more appropriate just to classify patients in three categories: a) those who have continued to have seizures and for how long; b) those who have resumed to have seizures after an interval that could be 1 year (arbitrary); c) those who had no longer have seizures at all.

It’s quite unusual for death to be an evolution of encephalitis. Could the Authors give us more information? What was the cause of death? Were they in the group of positive antibodies or negative antibodies? The case series dates to the 2000, era when diagnostic tools were not like the current ones and autoantibodies were not dosed. How many patients go back to that period? Could one have an idea of the stratification of cases per year?
